# What Do Differences between Alternating and Sequential Diadochokinetic Tasks Tell Us about the Development of Oromotor Skills? An Insight from Childhood to Adulthood

**DOI:** 10.3390/brainsci13040655

**Published:** 2023-04-13

**Authors:** Mónica Lancheros, Daniel Friedrichs, Marina Laganaro

**Affiliations:** 1Faculty of Psychology and Educational Science, University of Geneva, 1205 Geneva, Switzerland; 2Department of Computational Linguistics, University of Zurich, 8006 Zurich, Switzerland

**Keywords:** diadochokinesis, alternating motion rate tasks, sequential motion rate tasks, syllabic rate, development, motor planning, articulatory execution

## Abstract

Oral diadochokinetic (DDK) tasks are common research and clinical tools used to test oromotor skills across different age groups. They include alternating motion rate (AMR) and sequential motion rate (SMR) tasks. AMR tasks involve repeating a single syllable, whereas SMR tasks involve repeating varying syllables. DDK performance is mostly discussed regarding the increasing rates of AMR and SMR tasks from childhood to adulthood, although less attention is given to the performance differences between SMR and AMR tasks across age groups. Here, AMR and SMR syllabic rates were contrasted in three populations: 7–9-year-old children, 14–16-year-old adolescents and 20–30-year-old adults. The results revealed similar syllabic rates for the two DDK tasks in children, whereas adolescents and adults achieved faster SMR rates. Acoustic analyses showed similarities in prosodic features between AMR and SMR sequences and in anticipatory coarticulation in the SMR sequences in all age groups. However, a lower degree of coarticulation was observed in children relative to adults. Adolescents, on the contrary, showed an adult-like pattern. These findings suggest that SMR tasks may be more sensitive to age-related changes in oromotor skills than AMR tasks and that greater gestural overlap across varying syllables may be a factor in achieving higher rates in SMR tasks.

## 1. Introduction

Speaking is a complex oromotor skill that requires the smooth coordination of several muscles of the orofacial and laryngeal system. The ability to speak is usually learned during infancy and requires a number of years before being completely mastered. Gaining insights into children’s oromotor ability from an early age is rather challenging; however, a research and clinical tool that has been widely used for this purpose in populations of different ages, starting from infants, is the oral diadochokinetic (DDK) task test [1]. These tasks involve the rapid repetition of either a single syllable, commonly /pa/, /ta/ or /ka/, or a sequence of two or three different syllables, such as /pataka/, known respectively as alternating motion rate (AMR) and sequential motion rate (SMR) tasks. A difference in performance between AMR and SMR tasks has been consistently reported in adults, with faster rates for SMR relative to AMR; however, it remains unclear whether this difference is related to a mature (speech) motor control system or whether the AMR–SMR difference is already present in younger age groups. In the present study, we thus compare the production of AMR and SMR tasks from childhood to adulthood to investigate whether and how the performance of two DDK tasks vary across ages, with the ultimate goal of disentangling some hypotheses on the origin of the different AMR–SMR rates observed in adults.

DDK performance is usually measured by means of the speech rate [2], calculated by either counting the number of repeated syllables within a previously fixed interval or by measuring the time needed to repeat a given sequence [3]. Performance in DDK tasks has been largely tested across different age groups (e.g., in children [4,5,6,7,8], in adolescents [4,6,7,9], in young adults [5,10,11,12,13,14] and in the elderly [10,12,15,16]), with the most prominent result showing a gradual increase in the DDK rates from childhood to middle adulthood, followed by decreased rates from late adulthood. Although most DDK studies have commonly centered their discussion on those changes in rate over time, less attention has been paid to the performance pattern arising from the comparison between SMR and AMR tasks across different age groups. Additionally, many DDK studies on adults have surprisingly reported faster rates for SMR sequences relative to AMR syllables [5,12,13,14,17,18], which is a counterintuitive result since one would rather expect the repetition of a single syllable to reach faster rates.

Although the phenomenon of AMR tasks being slower compared with SMR tasks in adults has not received much attention, previous studies have consistently reported this pattern of performance between alternating tasks and their non-alternating counterparts. Some authors have attributed those results to the overlapping gestures allowed during the execution of SMR tasks [19], suggesting that they facilitate performance and contribute to an articulatory advantage over AMR tasks. For instance, in the case of producing the SMR sequence /bada/, articulatory overlapping between labial and lingual gestures is allowed, whereas the same is not possible during the repetition of the same syllable (i.e., /baba/ or /dada/). Bohland et al. [20] have proposed a different hypothesis on the slowing down of repeated syllables, this time at a higher processing level. They suggest that when selecting the motor program for a particular syllable (such as /pa/), the chosen program is actively inhibited right after selection in order to avoid reselection, slowing down the repetitive production of that same syllable. Although the authors did not specifically refer to the DDK tasks, the inhibition of a selected motor program can explain slower execution of AMR sequences (/baba/) relative to SMR sequences (/bada/).

The two hypotheses for faster rates in SMR relative to AMR tasks (i.e., overlapping gestures or motor program inhibition) relate to mechanisms of a mature motor speech encoding system, meaning that the differences between SMR and AMR might not be expected in childhood if we consider that coarticulation and other adult-like speech properties are achieved only adolescence (e.g., [21,22,23,24]). However, from previous studies it remains unclear whether the difference in performance between SMR and AMR tasks is already evident in younger age groups or whether it follows an evolutionary pattern from childhood to adulthood, as has been reported for the syllabic rates of SMR and AMR tasks separately. Changes in performance between AMR and SMR from childhood to adulthood can hardly be inferred from previous studies since SMR and AMR tasks are not systematically tested in DDK studies (e.g., in children and adolescents [25,26,27], in adults [28,29,30] or whenever they are included, no comments on the performance pattern between the two types of DDK tasks is found (e.g., in children and adolescents [7,31] or in adults [13,14]). Additionally, most studies are carried out only on one age group or only on one type of DDK—mostly AMR—which does not inform of the changes in performance between AMR and SMR from childhood to adulthood.

In a recent review by Kent et al. [32] on the oral diadochokinetic rates across different age groups, the authors calculated the means, medians and ranges per age group to get an overall lifespan pattern of DDK rates. Although the authors reported SMR and AMR rates separately and only commented on the general incremental pattern of SMR and AMR tasks from 2-year-olds until middle adulthood (and the following rate decline with advanced aging), a visual screening of the reported data (Table 13 in [32]) reveals additional information: SMR rates being higher than AMR rates does not seem to be a common finding across age groups. In fact, from the mean data on the age groups of interest, it is evident that: (1) children present with lower syllabic rates for SMR as compared with AMR tasks; (2) from the age group including 10-to-19-year-olds the DDK performance pattern switches, with SMR tasks achieving higher rates relative to the AMR syllable /ka/; and (3) subsequent age groups, from 20 to 50 years, reach higher SMR rates as compared with the three AMR syllables (e.g., /pa/, /ta/ and /ka/).

It is important to note that the averaged results reported by Kent et al. [32] are derived from different studies that do not necessarily report congruent findings. Nonetheless, the overall lifespan DDK rate does exhibit the SMR–AMR pattern described above. In children, for example, despite some studies reporting higher SMR rates (e.g., [5,7,33]), the reverted pattern is the most prominent in the DDK literature in childhood, with SMR performances yielding lower syllabic rates as compared with AMR tasks [4,6,9,31]. Concerning the adult data, the main finding across studies seems to be that SMR tasks reach higher syllabic rates compared with AMR tasks [5,12,13,14,17,18], even though some other studies report contradicting results for this age group [10,34,35]. Finally, DDK studies on adolescents do not point in any clear direction: half of the studies report lower rates for SMR tasks as compared with the fast repetition of the same syllable (e.g., [4,9,36]), whereas the other half finds the opposite pattern [6,7]. Importantly, DDK studies in this population are not only considerably scarcer compared with those in children or adults but they also include a narrower range of ages, mostly limited to young adolescents (e.g., between 11 and 13 years of age).

To gain a comprehensive understanding of the differences in performance between SMR and AMR tasks from childhood to adulthood, the present study investigates the production of the same SMR and AMR sequences in three different age groups: 7–9-year-old children, 14–16-year-old adolescents and 20–30-year-old adults. We expect to replicate the prominent results previously reported in adult DDK studies, namely, higher SMR rates compared with the repetitive production of the same syllable. Concerning children, we expect no differences between SMR and AMR if they rely on an immature motor speech encoding system at the ages of 7–9. As for adolescents, there are three plausible hypotheses: (1) differences between SMR and AMR tasks would be absent in 14–16-year-old adolescents if they continue to rely on an immature motor speech encoding system; (2) SMR and AMR tasks would differ in some aspects if adolescents’ speech motor control system is nearly adult-like; and (3) 14–16-year-old adolescents would display differences between SMR and AMR tasks if their motor speech encoding system is fully mature, mirroring DDK performances in adults.

To further analyze whether SMR achieving faster rates due to the chunking of the three /badego/ syllables [37], we conducted an acoustic analysis comparing the prosodic pattern between SMR and AMR syllables. This analysis was performed separately for each age group. We also measured anticipatory coarticulation across groups to test the gestural overlap proposed by Ziegler et al. [19].

## 2. Materials and Methods

### 2.1. Participants

The study involved 71 participants who were divided into three age groups: children (n = 24, 11 males, mean age: 8 years, range: 7–9 years), adolescents (n = 22, 10 males, mean age: 15 years, range: 14–16 years) and young adults (n = 25, 11 males, mean age: 25 years, range: 20–30 years). All participants were French native speakers or acquired French before the age of six, with no reported hearing, language, speech, neurologic or psychiatric disorders. Participants or their legal representatives (in the case of children) provided informed consent to be included in the study. Only children and adolescents were paid for their participation since they were involved in a different research protocol. The study was approved by the local ethics committee of the University of Geneva.

### 2.2. Material

The study employed the DDK tasks from the French speech assessment protocol MonPaGe [38,39]. The tasks included three AMR sequences (/ba/, /de/ and /go/) and one SMR production (/badego/). The three CV syllables targeted different articulators: The jaw /lips with /ba/, the front part of the tongue with /de/ and the body of the tongue with /go/. The syllables also differed in place of articulation (bilabial, alveolar and velar, respectively) and in oral frequency values in French: 1986.31, 1078.9 and 468.52 occurrences per million syllables, respectively, according to the database “LEXIQUE” [40,41].

### 2.3. Procedure

Before the production of the DDK tasks, participants were shown a visual and auditory example of the fast repetition of the syllable /ta/ to familiarize themselves with the required task. Thus, while “tatata…” was written on the screen, participants listened to the recorded example of the syllable /ta/, produced at a syllabic rate of 5.48. They were then presented with each sequence of syllables and asked to repeat it as fast and accurately as possible on a single breath for at least five seconds. In each trial, participants were first asked to read the presented sentence aloud to ensure they correctly identified and produced the target sequence (e.g., “bababa” was presented on the screen and participants had to pronounce the syllable /ba/ three times at a normal rate before starting the DDK task). The stimuli included in the DDK task were always presented in the same order: each of the three AMR tasks /ba/, /de/ and /go/ was presented on the screen, followed by the presentation of the SMR task /badego/. Experimental runs were audio recorded separately per sequence. Participants were allowed to restart their productions whenever they failed to perform the DDK task.

Audio recordings were made with a cardioid Shure SM58 microphone (SHURE Inc., Niles, IL, USA) and PreSonus Audiobox USB audio interface (PreSonus Audio Electronics, Inc., Baton Rouge, LA, USA) with 24-bit resolution at a sampling rate of 48 kHz. Participants maintained a constant distance from the microphone during the recordings.

### 2.4. Syllabic Rate Analysis

The consonant and vowel segments were labeled manually in Praat [42] in order to use a custom Praat script that calculates the syllabic rate (number of syllables per second) in approximately the first four seconds of each DDK production. The analysis window started from the zero-crossing onset of the waveform and was hand-corrected to include the last syllable—or sequence of syllables in the case of /badego/—within this time frame. The syllabic rate corresponded to the number of syllables produced per second within the analyzed time-window for each individual production.

For data cleaning purposes, rates exceeding 2.5 standard deviations from each type of task’s mean per group were excluded. From the children group, one participant was excluded since all his AMR rates were considered outliers (final number of children = 23). From the group of 14-to-16-year-old adolescents, apart from one instance for the AMR rate of the syllable /ba/ not being included due to recording problems for one participant, a second /ba/ rate value from another adolescent was excluded after cleaning the data. From the adult data, one adult’s syllabic rate for the AMR syllable /de/ was also discarded following the cleaning procedure.

Syllable rate data were fitted with mixed models [43] with RStudio software 2022.02.3 + Build 492 (R-project, R-development core team 2005) in order to compare the rates of the two DDK tasks (i.e., SMR vs. AMR) across groups (i.e., children, adolescents and adults).

### 2.5. Acoustic Analyses

#### 2.5.1. Prosodic Features

Fundamental frequency (*f*_o_) contours and intensity curves of the AMR and SMR sequences were generated for the first twelve CV repetitions using ProsodyPro (version 5.7.8.7, [44]), implemented in Praat. The sampling rate for the extraction was set to 100 Hz, and the results were manually corrected. Prosodic measures were analyzed to investigate whether participants used any of these features to realize prosodic boundaries (i.e., chunking) in the AMR and SMR sequences. *f*_o_ and intensity measurements were time-normalized by dividing each CV interval into the same number of data points (N = 10) for better comparability and illustration purposes.

Importantly, in a sequence of three /ba/ productions (i.e., /bababa/), only the first AMR /ba/ syllables were compared with the corresponding SMR /ba/ syllables in /badego/. The same procedure was applied to the AMR /de/ and AMR /go/ syllables (i.e., the second AMR /de/ productions were compared with the SMR /de/ syllables and the third AMR /go/ productions with the SMR /go/ syllables) to ensure comparable values across the same vowels, especially for the *f*_o_. To test whether SMR sequences achieve faster rates due to sequence chunking, we contrasted the *f*_o_ and the intensity peaks of the syllable /go/ between AMR and SMR tasks across groups. Only the syllable /go/ was chosen here because, in French, the overall *f*_o_ and intensity contours are well-known to rise on the last syllable of the sequence [45].

#### 2.5.2. Anticipatory Coarticulation

The distribution of the first two formant frequencies (F1 and F2) was measured in the AMR and SMR sequences with custom Praat scripts using linear predictive coding (LPC) with variable-order prediction filters (LPC order ranging from 12 to 16) to take into account the different vocal tract sizes of children, adolescents and adults. Specifically, we calculated a compactness index between F1 and F2 for the /a/ vowels in the AMR and SMR tasks, which indicates the degree to which the F1–F2 space of one vowel moves towards another vowel category due to anticipatory coarticulation. To obtain this index, we subtracted the single F2–F1 values obtained in the middle portion of the SMR and AMR vowels /a/. The other SMR vowels (i.e., /e/ and /o/) were not statistically tested.

## 3. Results

Raw data are available in the Appendix A.

### 3.1. Syllabic Rate

Mean AMR and SMR syllabic rates in syllables per second are shown for the three groups in Table 1.

The linear mixed model (Model: lmer(Rate~(task*group) + (1|subject), data = data, REML = FALSE) indicated a main effect of type of task (F(1, 206.65) = 46.61; *p* < 0.001) and of group (F(2, 75.92) = 20.85; *p* < 0.001) as well as an interaction between the two variables (F(2, 206.65) = 8.30; *p* < 0.001). Interaction contrasts, examined using the package “emmeans” [46] and corrected for multiple comparisons using Tukey’s method, revealed no difference between SMR and AMR tasks for children (t(210) = −1.61; *p* = 0.11, β = −0.19, SE = 0.12), whereas a significant difference between the two types of DDK tasks was found for adolescents (t(210) = −2.99; *p* < 0.01, β = −0.37, SE = 0.12) and adults (t(210) = −7.29; *p* < 0.001, β = −0.84, SE = 0.12).

Given those results, and especially considering the smaller estimate value for adolescents as compared with that of adults, we investigated whether differences between SMR and AMR in adolescents were driven by some (but not all) of the AMR syllables. Thus, separate models for those two age groups were performed by setting the type of sequence (/ba/, /de/, /go/ and /badego/) as fixed factor. Since we were still interested in the contrast between SMR and the three AMR syllables, /badego/ was always set as the intercept in contrasts.

#### Additional Syllabic Rate Analysis

Adolescents

Mean syllabic rates per type of sequence in adolescents are presented in Figure 1.

The linear mixed model on the 22 adolescents yielded a main effect of type of sequence (F(3, 64.19) = 14.95; *p* < 0.001). Contrasts revealed significantly faster syllabic rates for the SMR task as compared with the AMR syllables /de/ (t(64.06) = −3.12; *p* < 0.003, β = −0.30, SE = 0.09) and /go/ (t(64.06) = −6.43; *p* < 0.001, β = −0.62, SE = 0.09). The comparison between the SMR syllable and the AMR syllable /ba/ did not reach significance (*p* > 0.05).

Adults

Adults’ mean syllabic rates per type of sequence are shown in Figure 2.

The linear mixed model on the adult data indicated a main effect of type of sequence (F(3, 73.82) = 19.34; *p* < 0.001). Contrasts revealed significantly faster syllabic rates for the SMR task as compared with all the AMR syllables: /ba/ (t(73.76) = −4.44; *p* < 0.001, β = −0.65, SE = 0.15), /de/ (t(73.88) = −5.15; *p* < 0.003, β = −0.77, SE = 0.15) and /go/ (t(73.76) = −7.42; *p* < 0.001, β = −1.09, SE = 0.15).

### 3.2. Acoustic Analyses

Twelve speakers (7 children, 3 adolescents and 2 adults) were excluded from the acoustic analyses due to noise in the audio signal.

#### Prosodic Features

Fundamental frequency (*f*_o_)

Figure 3 shows *f*_o_ contours for the averaged /ba/, /de/ and /go/ syllables from the AMR and SMR tasks per age group.

*f*_o_ data were fitted with mixed models (Model: lmer(F0~(DDK_task*group) + (1|Subject), data = data, REML = FALSE), revealing a main effect of group (F(2, 59) = 18.99; *p* < 0.001), with children showing higher *f*_o_ values as compared with both adolescents and adults, whereas no main effect of task (F(1, 413) = 0.47; *p* = 0.49) and no interaction between the two factors was found (F(2, 413) = 2.01; *p* = 0.13). These results suggest no difference in *f*_o_ between AMR and SMR tasks for children, adolescents and adults.

Intensity

Figure 4 shows intensity contours (in dB SPL) for the averaged /ba/, /de/ and /go/ syllables from the AMR and SMR tasks per age group.

Mixed models (Model: lmer(Intensity~(DDK_task*group) + (1|Subject), data = data, REML = FALSE) yielded no main effect of group (F(2, 59) = 0.02; *p* = 0.97) or task (F(1, 413) = 1.18; *p* = 0.27). The interaction between group and task was neither significant (F(2, 413) = 0.23; *p* = 0.79). Therefore, the intensity results show no difference between AMR and SMR tasks for children, adolescents and adults, which is consistent with the *f*_o_ findings reported above.

### 3.3. Anticipatory Coarticulation

The F2–F1 space for the SMR and AMR /a/ vowels is shown in Figure 5 for each age group. SMR vowels /e/ and /o/ are also included in the graphs although they were not statistically contrasted.

Mixed models on the F2−F1 distance of the AMR and SMR vowel /a/ (Model: lmer(F2F1~(DDK_task*group) + (1|Subject), data = data, REML = FALSE) revealed a main effect of group (F(2, 61.27) = 9.01; *p* < 0.001), with children having higher values compared with adolescents and adults, as well as a main effect of task (F(1, 885) = 831.01; *p* < 0.001), indicating more F2/F1 distance for the SMR vowel /a/. The interaction between group and task similarly reached significance (F(2, 885) = 6.93; *p* < 0.001). Relevant contrasts showed that the F2–F1 distance of the SMR vowel /a/ was significantly increased in the three groups (children: t(888) = −13.15, *p* < 0.001, β = −261, SE = 19.8; adolescents: t(888) = −16.41, *p* < 0.001, β = −308, SE = 18.8; adults: t(888) = −20.97, *p* < 0.001, β = −358, SE = 17.1), confirming gestural overlap for SMR sequences in children, adolescents and adults. However, by visual inspection of Figure 5, the degree of anticipatory coarticulation seemed to differ across groups from the formant trajectories in the normalized vowel space (F2/F1, see Figure 6), with adolescents and adults showing formant transitions in the vowel /a/ towards the values of the following vowel /e/ in the SMR sequences. This suggests that anticipatory coarticulation might be weaker in children relative to adolescents and adults.

In order to determine whether the degree of anticipatory coarticulation differed between groups, additional intergroup analyses were conducted. To account for potential errors in formant measurements due to higher *f*_o_ and the resulting sparser distribution of the harmonics (which makes estimations for children a more error-prone process), a whole-spectrum measure of coarticulation was calculated. This was obtained by the Euclidean distance between average Mel-frequency spectral vectors of the /a/ vowels from the SMR and AMR tasks. Mixed models were used to statistically test the Euclidean distance between the SMR and the AMR /a/ vowels across groups (Model: lmer(EDistance~(group) + (1|Subject), data = data, REML = FALSE). The results revealed a main effect of group (F(2, 59) = 4.21; *p* < 0.03), with children obtaining lower values compared with adolescents (t(59) = 2.11; *p* = 0.04, β = 4.81, SE = 2.27) and with adults (t(59) = 2.82; *p* < 0.01, β = 6.15, SE = 2.18). No differences between adolescents and adults were found (t(59) = −0.63; *p* = 0.53, β = −1.34, SE = 2.11). These results (see Figure 6) suggest that the degree of anticipatory coarticulation is significantly lower for children, whereas 14-to-16-year-old adolescents coarticulate SMR sequences to the same extent as adults. (Since children’s acoustic data included a smaller sample compared with the adolescent and adult groups, the Euclidian distance values were run on a dataset of 17 participants per group. Thus, 6 adults and 2 adolescents were randomly removed in order to reach the same number of participants across groups. Mixed models indicated the same results, with children presenting a lower degree of anticipatory coarticulation compared with adults, whereas no difference between adolescents and adults was found). 

## 4. Discussion

In the present study, we compared the performance of SMR and AMR tasks across three different age groups—children, adolescents and adults—with the aims of better understanding oromotor skill development and of shedding light on some hypotheses about the mechanisms underlying the different rates for SMR and AMR in adults. Our findings suggest that DDK rates for alternating and non-alternating tasks were similar in children, whereas distinct rate patterns between SMR and AMR tasks were observed in adolescents and adults. Specifically, adolescents exhibited higher SMR rates in comparison with the AMR syllables /de/ and /go/, whereas young adults presented higher SMR rates compared with all AMR syllables (i.e., /ba/, /de/ and /go/). Thus, contrasting SMR and AMR rates across the three age groups revealed that SMR and AMR tasks follow different performance patterns from childhood to adulthood.

Producing repetitions of the same syllable has been previously suggested to differ from the repetitive production of changing syllables [47]. Experimental studies have also reported diverging results in the repetition performances between SMR and AMR tasks in both neurotypical participants of different ages and in pathological populations (e.g., in healthy participants [5,12] or in Parkinson’s disease or patients with traumatic brain injury [48,49]). However, to the best of our knowledge, no studies have compared the performance pattern of the two DDK tasks across different age groups. Notably, the few studies that have investigated DDK rates including both SMR and AMR tasks did not always calculate directly comparable measurements between tasks. For instance, Modolo et al. [31], who investigated SMR and AMR rates in 8-to-10-year-old Brazilian-Portuguese-speaking children, calculated the AMR rates by counting the number of /pa/, /ta/ and /ka/ syllables produced in a given time, whereas the SMR rates were calculated by counting the number of /pataka/ chunks in a given time.

The present results clearly show that DDK tasks are characterized by a gradual distinction between SMR and AMR tasks over time, thus also confirming that DDK tasks are a heterogeneous class of motor behavior [47]. To better understand why SMR tasks achieve higher syllabic rates in 14-to-16-year-old adolescents and in adults, we investigated the two explanatory hypotheses presented in the Introduction by conducting acoustic analyses. The first hypothesis, proposing SMR tasks being faster because of the chunking of their different syllables (here /badego/, [37]), was tested by analyzing prosodic features (fo and intensity) on the last syllable of the SMR sequence (/go/) as compared with the AMR syllable /go/ resulting from the last repetition of each successive three-syllable sequence. The results showed no significant difference in the two prosodic features between SMR and AMR tasks, indicating that the first hypothesis was not supported. In other words, the higher syllabic rates found in the older groups for SMR tasks cannot be explained by the integration of separate movement elements into ‘‘chunks’’.

The second hypothesis explaining the faster production of SMR sequences relies on higher gestural overlap across SMR syllables [19]. This hypothesis was investigated by exploring the amount of anticipatory coarticulation in the SMR sequences. The results suggest anticipatory coarticulation in the SMR sequences in all three age groups, which is in line with the hypothesis of sequential tasks allowing gestural overlap [19] but contradicts the syllabic rate findings showing faster SMR rates only for adolescents and adults. However, additional analysis across age groups showed a reduced degree of coarticulation in children compared with adults, whereas 14-to-16-year-old adolescents presented with an adult-like pattern of coarticulation for the SMR sequences. Taken together, these findings suggest that the two older groups might achieve higher SMR rates because of the greater gestural overlap across alternating syllables.

Finally, although this study adopted an acoustic perspective for investigating why SMR tasks achieve faster rates than their non-alternating counterparts, it is important to acknowledge that higher-level mechanisms might also play a role in preventing gestural overlap across repeated syllables (i.e., AMR tasks). For instance, motor programming inhibition [20] may play a role in preventing gestural overlap across repeated syllables in AMR tasks, which could also explain the slower rates observed in this task in older groups. In fact, high- and low-level hypotheses are not mutually exclusive since they touch different processing levels that could coexist. For instance, it is reasonable to conceive that adolescents and adults reach faster SMR rates thanks to the gestural overlap allowed across alternating syllables, while in parallel their AMR rates might be slowed down due to the motor program inhibition of repeated syllables. This, however, needs to be experimentally tested in future studies. Investigating the differences in brain activation during the production of SMR and AMR tasks with neuroimaging techniques, such as magneto/electroencephalography (M-EEG), might be pertinent to test the higher-level hypothesis of motor programming differences between the two DDK tasks.

Although this study included three different groups, testing the SMR–AMR performance patterns in other intermediate ages, especially in speakers from 9 to 12 years of age, is needed in future studies. Including intermediate groups would allow determination of whether the changes observed in 14-to-16-year-old adolescents follow a linear and progressive development from childhood or if a crucial switching change can be identified at a particular age. Inclusion of larger sample sizes is also encouraged in future investigations to strengthen the statistical findings, particularly in younger groups where larger between-participant variability is generally observed.

The results of the present study bring important insights into the development of oromotor skills throughout a speech-like task that is easily used across different age groups. We were able to show that the production of SMR sequences becomes faster than that of AMR sequences only in mid-adolescence and that this finding seems to be related, at least partially, to greater gestural overlap across alternating syllables. A developmental pattern of intra-gestural synergies reaching adult-like patterns only in adolescence has been previously reported for speech stimuli in acoustic and articulatory studies on speech production (e.g., [22]). The fact that the same pattern found for speech is mirrored in DDK tasks, despite their differences in terms of repetition and maximal-rate requirements [19], indicates that DDK tasks might bring relevant insights into the speech motor control state across different age groups.

## 5. Conclusions

In conclusion, the production of SMR and AMR sequences showed similar rates in children, whereas faster SMR rates were displayed in adolescents and adults as compared with AMR rates. Children achieving similar rates for the two DDK tasks seems to be due, at least partially, to a lower degree of coarticulation in the production of sequences of different syllables. The finding that faster SMR sequences are only attainable in mid-adolescence aligns with research on speech production, implying that DDK tasks could potentially provide valuable information on the state of speech motor control during development.

## Figures and Tables

**Figure 1 brainsci-13-00655-f001:**
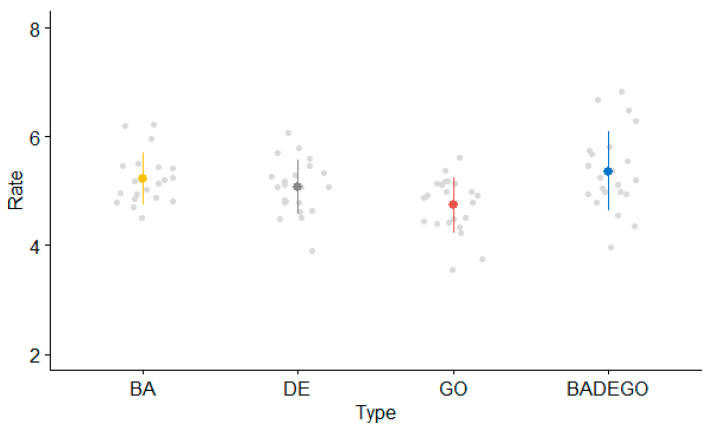
Adolescents’ mean syllabic rate (syllables per second) per type of sequence.

**Figure 2 brainsci-13-00655-f002:**
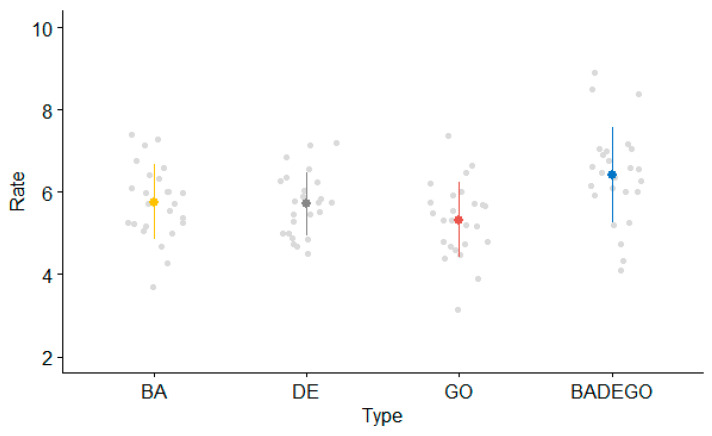
Adults’ mean syllabic rate (syllables per second) per type of sequence.

**Figure 3 brainsci-13-00655-f003:**
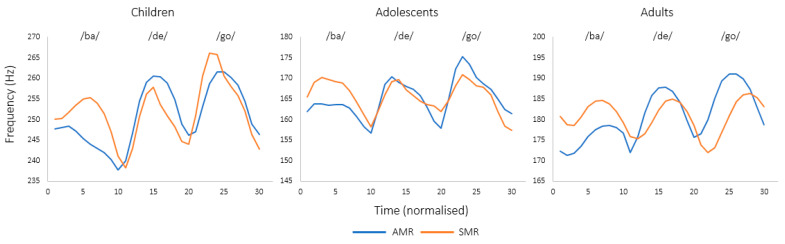
*f*_o_ contours for the averaged /ba/, /de/ and /go/ syllables from the AMR and SMR tasks in children, adolescents and adults.

**Figure 4 brainsci-13-00655-f004:**
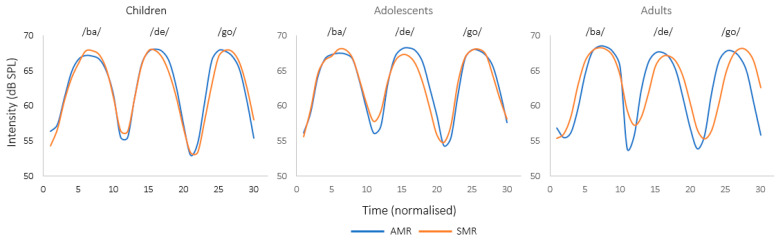
Intensity curves (in dB SPL) for the averaged /ba/, /de/ and /go/ syllables from the AMR and SMR tasks in children, adolescents and adults.

**Figure 5 brainsci-13-00655-f005:**
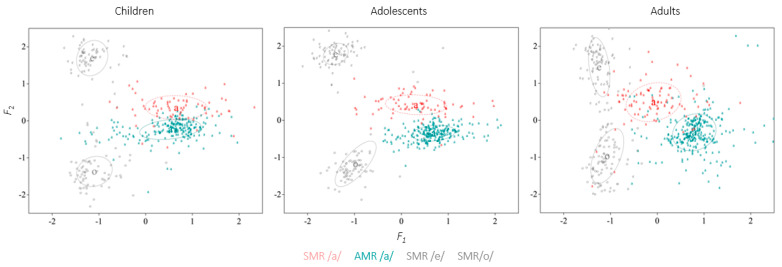
F2−F1 space for the SMR and AMR /a/ vowels (colored) across age groups.

**Figure 6 brainsci-13-00655-f006:**
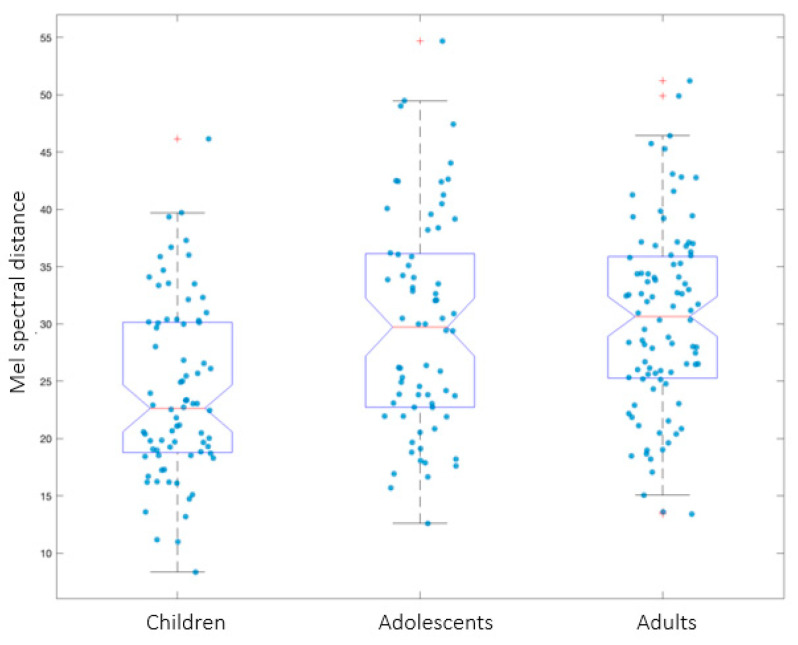
Mel spectral distance between the SMR and AMR /a/ across age groups. The data points represented by “+” correspond to outliers.

**Table 1 brainsci-13-00655-t001:** Mean SMR and AMR syllabic rates in syllables per second (SD) for children, adolescents and adults.

	Mean Syllabic Rates (SD)		Difference SMR–AMR (%)
	AMR	SMR	
Children	4.66 (0.64)	4.85 (1.06)	4%
Adolescents	5.00 (0.53)	5.37 (0.73)	7%
Adults	5.60 (0.88)	6.41 (1.17)	13%

## Data Availability

The raw data is available in the Appendix A.

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
