# Peer review of "What Do Differences between Alternating and Sequential Diadochokinetic Tasks Tell Us about the Development of Oromotor Skills? An Insight from Childhood to Adulthood"

_brainsci, 2023, doi:10.3390/brainsci13040655_

Round 1

Reviewer 1 Report

Thank you for sharing your paper titled "What do differences between alternating and sequential diadochokinetic tasks tell us about the speech motor control system? An insight from childhood to adulthood." After reviewing your manuscript, I do have some suggestions to enhance the paper.

Firstly, I appreciate the excellent presentation of your study and the quality of the design. However, to make your study more complete, I recommend that you add a conclusion, limitations, implications and recommendations sections. The conclusion should summarize the main findings and their significance. The limitations section should identify the limitations of the study and discuss how these limitations could affect the interpretation of the results. The implications section should discuss the implications of the study for future research and practice. Finally, the recommendations section should provide specific recommendations for future research and practice based on the study's findings.

Thank you for your hard work and dedication in conducting this study, and I look forward to seeing the final version of your manuscript.

Best regards,

Author Response

We thank you for your feedback and for your kind words. A conclusion paragraph was already added at the very end of the manuscript: we change it now in order to include a summary of the main findings as well as their significance. Concerning the recommendations about future research, the idea that was already stated in the previous version was expanded and some other suggestions were added. Limitations were coupled with the recommendations. Finally, some implications of the study were commented.

Reviewer 2 Report

The study presented in the manuscript is devoted to a crucial up-to-date problem - the mechanisms of speech motor control system. The main point of the manuscript is the motion rate of the alternating and sequential articulatory movements. The authors presented novel data about age-related changes in motion rate and some coarticulation measures. The text of the manuscript is well-structured, the data is statistically validated. However, some remarks should be stated:

1.       The passage in lines 213-224  should be better moved to the Method section.

2.       Despite the announced in the Header intention to tell “about the speech motor control system’’, very few mentions, comments, and inferences related to speech mechanisms were presented in the text. The system level of the speech motor processing and the modern speech motor programming and processing models were not discussed. In this regard, it seems reasonable to edit the Header to make it more relevant to the text content.

Author Response

1. The passage in lines 213-224 should be better moved to the Method section.

--> Thank you for pointing this out. The passage has been moved to the method section, as suggested.

2. Despite the announced in the Header intention to tell “about the speech motor control system’’, very few mentions, comments, and inferences related to speech mechanisms were presented in the text. The system level of the speech motor processing and the modern speech motor programming and processing models were not discussed. In this regard, it seems reasonable to edit the Header to make it more relevant to the text content.

--> The header is now changed in the new version: What do differences between alternating and sequential diadochokinetic tasks tell us about the development of oromotor skills? An insight from childhood to adulthood.

We thank you for this suggestion.